# Investigation of SARS-CoV-2 infection in dogs and cats of humans diagnosed with COVID-19 in Rio de Janeiro, Brazil

Guilherme Amaral Calvet[1]◉*, Sandro Antonio Pereira[2]◉, Maria Ogrzewalska[3]◉, Alex Pauvolid-Corrêa[3,4,5], Paola Cristina Resende[3,4], Wagner de Souza Tassinari[6], Anielle de Pina Costa[1], Lucas Oliveira Keidel[2], Alice Sampaio Barreto da Rocha[3,4], Michele Fernanda Borges da Silva[1], Shanna Araujo dos Santos[2], Ana Beatriz Machado Lima[3,4], Isabella Campos Vargas de Moraes[1], Artur Augusto Velho Mendes Junior[2], Thiago das Chagas Souza[3,4], Ezequias Batista Martins[1], Renato Orsini Ornellas[2], Maria Lopes Corrêa[2], Isabela Maria da Silva Antonio[2], Lusiele Guaraldo[1], Fernando do Couto Motta[3,4], Patrícia Brasil[1], Marilda Mendonça Siqueira[3,4‡], Isabella Dib Ferreira Gremião[2‡], Rodrigo Caldas Menezes[2‡]

**1** Acute Febrile Illnesses Laboratory, Evandro Chagas National Institute of Infectious Diseases, Oswaldo Cruz Foundation, Rio de Janeiro, Brazil, **2** Laboratory of Clinical Research on Dermatozoonoses in Domestic Animals, Evandro Chagas National Institute of Infectious Diseases, Oswaldo Cruz Foundation, Rio de Janeiro, Brazil, **3** Laboratory of Respiratory Viruses and Measles, Oswaldo Cruz Institute, Oswaldo Cruz Foundation, Rio de Janeiro, Brazil, **4** SARS-CoV-2 National Reference Laboratory for the Brazilian Ministry of Health and Regional Reference Laboratory in Americas for the Pan-American Health Organization, Brazil, **5** Department of Veterinary Integrative Biosciences, Texas A&M University, College Station, Texas, United States of America, **6** Mathematics Department, Exact Sciences Institute, Federal Rural University of Rio de Janeiro, Rio de Janeiro, Brazil

◉ These authors contributed equally to this work.
‡ These authors also contributed equally to this work.
* guilherme.calvet@ini.fiocruz.br

## Abstract

### Background

Infection by SARS-CoV-2 in domestic animals has been related to close contact with humans diagnosed with COVID-19. Objectives: To assess the exposure, infection, and persistence by SARS-CoV-2 of dogs and cats living in the same households of humans that tested positive for SARS-CoV-2, and to investigate clinical and laboratory alterations associated with animal infection.

### Methods

Animals living with COVID-19 patients were longitudinally followed and had nasopharyngeal/oropharyngeal and rectal swabs collected and tested for SARS-CoV-2. Additionally, blood samples were collected for laboratory analysis, and plaque reduction neutralization test (PRNT$_{90}$) to investigate specific SARS-CoV-2 antibodies.

### Results

Between May and October 2020, 39 pets (29 dogs and 10 cats) of 21 patients were investigated. Nine dogs (31%) and four cats (40%) from 10 (47.6%) households were infected with

**Data Availability Statement:** Data underlying the study cannot be made publicly available due to

ethical concerns, as data contain several personally identifiable information. Data are available from Oswaldo Cruz Foundation for researchers who meet the criteria for access to confidential data. Contact information: Institutional Ethics and Research Committee of the Evandro Chagas National Institute of Infectious Diseases, email: cep@ini.fiocruz.br or Guilherme Amaral Calvet; email: guilherme.calvet@ini.fiocruz.br.

**Funding:** MMS: This study was supported by CGLab/MoH (General Laboratories Coordination of Brazilian Ministry of Health), CVSLR/FIOCRUZ (Coordination of Health Surveillance and Reference Laboratories of Oswaldo Cruz Foundation), The National Council for Scientific and Technological Development (CNPq) COVID-19 MCTI 402457/ 2020-0, INOVA VPPCB-005-FIO-20-2-69, and Carlos Chagas Filho Foundation for Research Support of the State of Rio de Janeiro (FAPERJ) E26/210.196/2020.

**Competing interests:** The authors have declared that no competing interests exist.

or seropositive for SARS-CoV-2. Animals tested positive from 11 to 51 days after the human index COVID-19 case onset of symptoms. Three dogs tested positive twice within 14, 30, and 31 days apart. SARS-CoV-2 neutralizing antibodies were detected in one dog (3.4%) and two cats (20%). In this study, six out of thirteen animals either infected with or seropositive for SARS-CoV-2 have developed mild but reversible signs of the disease. Using logistic regression analysis, neutering, and sharing bed with the ill owner were associated with pet infection.

## Conclusions

The presence and persistence of SARS-CoV-2 infection have been identified in dogs and cats from households with human COVID-19 cases in Rio de Janeiro, Brazil. People with COVID-19 should avoid close contact with their pets during the time of their illness.

## Introduction

In the current coronavirus disease (COVID-19) pandemic caused by severe acute respiratory syndrome coronavirus 2 (SARS-CoV-2), there are significant gaps in understanding the role of vertebrates in their transmission and whether there are intermediate hosts that can act as reservoirs and/or amplifying hosts. It is believed that SARS-CoV-2 was originated in wild animals and, later, transmitted to humans [1]. A study showed that SARS-CoV-2 was 96% identical to a coronavirus found in bats, suggesting that these animals would be reservoirs of the ancestor of this virus [2]. However, the intermediate hosts that caused the transmission to humans are still unknown [3]. The investigation of intermediate hosts of SARS-CoV-2 can help to understand the dynamics of COVID-19 and to evaluate the possibility of zoonotic transmission. Recent findings suggested that the infection in minks can result in spillover back into humans [4]. The infection of animals with the SARS-CoV-2 virus may have implications for animal health and welfare, wildlife conservation, and biomedical research [4].

Studies have shown that some wildlife and domestic animals can be naturally or experimentally infected with SARS-CoV-2 [5]. However, acute infection in cats or dogs has been less reported [6, 7]. Few studies have demonstrated that SARS-CoV-2 infection in dogs and cats is mostly detected in animals living in households with at least one SARS-CoV-2-infected human, suggesting that the transmission may have occurred from humans to pets [8–16]. However, epidemiological investigations and longitudinal studies are limited and involve a small number of animals [8, 9, 11, 15, 17]. Therefore, it is not yet known how often this infection occurs, as well as whether animals develop clinical signs and whether there is a zoonotic and animal transmission of this virus under natural conditions.

In Brazil, there have been almost nine million confirmed COVID-19 cases and 214 thousand deaths by January 22, 2021 [18]. The state of Rio de Janeiro (RJ) located in the Southeast of Brazil is one of the most affected states in the country [18]. So far, only five cases of SARS-CoV-2 infection in cats and six cases in dogs [16, 19, 20] were reported in Brazil, and none of them were from RJ.

The main objectives of this study were 1) To assess the exposure, infection, and persistence of SARS-CoV-2 in nasopharyngeal and oropharyngeal secretions (nasal and oral swabs) and feces (rectal swab) of dogs and cats living in households with a COVID-19 human case; 2) To evaluate risk factors associated with SARS-CoV-2 infection in pets.

## Material and methods

### Study human component

**Setting and inclusion criteria.** The human component of the study was conducted at the Acute Febrile Illness outpatient clinic at the Evandro Chagas National Institute of Infectious Diseases (INI), Oswaldo Cruz Foundation (Fiocruz), in Rio de Janeiro, Brazil, and the patients' residences. The index cases were male or female patients aged 18 years and older, with COVID-19 confirmed by real-time reverse transcription polymerase chain reaction (real-time RT-PCR) performed in nasopharyngeal/oropharyngeal (NP/OP) swabs and/or unstimulated whole saliva (UWS). Household contacts of index cases were tested and offered to participate in the study including teenagers and children, because they may likely share a bedroom with their pets and enhance potential transmission.

**Investigation of human COVID-19 infection and follow-up visits.** A systematic syndromic investigation utilizing a specific laboratory algorithm, which included diagnostics assays for detection of COVID-19 cases was performed in all patients with suspected SARS-CoV-2 infection and their household contacts. Clinically suspected COVID-19 cases were defined as the presence of fever or chills, cough, shortness of breath or difficulty breathing, fatigue, muscle or body aches, headache, the new loss of taste or smell, sore throat, congestion or runny nose, nausea or vomiting, and diarrhea [21]. Clinically suspected COVID-19 patients were qualified to be included and followed-up in the study if they were tested positive for SARS-CoV-2 by real-time RT-PCR and cohabit with dogs and or cats. Data were collected and managed using a Research Electronic Data Capture (REDCap) tool hosted at the study center. Subsequentially, information about sociodemographic features, epidemiological characteristics, clinical signs and symptoms, the severity of the disease, number of household contacts, animal habits, and laboratory test results were compiled. Also, written informed consent for human and pet sampling was obtained from all patients. At the first and in two follow-up visits done at 15 (± 3 days) and 30 (± 7 days) after the first visit, the following procedures were carried out: 1) Informed consent process (first visit only); 2) Detailed questionnaire (REDCap-based CRF survey completion); 3) Collection of UWS and two NP and one OP swabs. Combined NP/OP swabs were placed in a single tube with 3 mL of Viral Transport Medium (VTM). VTM containing cell culture medium, antibiotics, antimycotic, and fetal bovine serum. The samples kept on the ice were delivered on the same day to the Regional Reference Laboratory in the Americas for Coronavirus.

**Detection of SARS-CoV-2 in human samples.** Nucleic acids were extracted from VTM samples by the QIAamp Viral RNA Mini kit (Qiagen), or by Chemagic 360 (Perkin-Elmer). Extracted RNA was submitted to a SARS-CoV-2 real-time RT–PCR by the SARS-CoV-2 detection Molecular E/RP Kit (Biomanguinhos, Rio de Janeiro, Brazil) according to Corman et al. [22]. Reverse transcription and amplification were conducted in the ABI7500 platform.

### Study animal component

**Setting and inclusion criteria.** The animal part of the study was conducted by veterinarians of the Laboratory of Clinical Research on Dermatozoonoses in Domestic Animals (Lapclin-Dermzoo), INI, Fiocruz. Only dogs and/or cats living in the same household of human patients with confirmed COVID-19 were included in the study. The cats aged less than 12 months old or more than ten-years-old were not included in the study because of sedation concerns. Pregnancy was also an exclusion criterion. During the first animal visit, another consent form was obtained from the pets' owners to enroll their animal (s) in the study. The consented form included acknowledgment for clinical examination, sedation, and sample collection.

**Investigation of animal SARS-CoV-2 infection, coinfections, and follow-up visits.**
*Clinical examination.* For the clinical examination, an appropriate muzzle was applied to the dogs, and both cats and dogs were physically restrained. The clinical examination consisted of the following procedures: general inspection, evaluation of health condition (good, fair, and poor) and the degree of hydration, an inspection of the skin and mucous membranes (oral, conjunctive, genital, and anal), palpation of lymph nodes (mandibular, parotid, axillary and popliteal) and abdominal organs, cardiopulmonary auscultation, and rectal temperature measurement. The dogs were physically restrained to collect the biological samples. The cats were sedated by an intramuscular injection of a combination of 10% ketamine hydrochloride (10 mg/kg) and 1% of acepromazine maleate (0.1 mg/kg).

*Specimen collection.* The blood samples were collected by cephalic or jugular vein venipuncture. A total of 3–5 mL of blood were collected and packed in tubes with EDTA anticoagulant for a complete blood count using automated equipment (Sysmex Poch-100iv Diff $^{TM}$, Sysmex, Japan). Also, 2–3 mL of blood were filled into tubes without anticoagulant to obtain serum for $PRNT_{90}$ and biochemistry analysis using automated equipment (Bioclin 3000, Bioclin, Brazil). The sampling of nasopharyngeal and oropharyngeal secretions and feces was preferably performed before the ingestion of water and food. A combination of one oral/oropharyngeal swab and two nasal/nasopharyngeal swabs were used as a respiratory sample. All three swabs were placed together in a single 15 mL sterile tube containing 3 mL of VTM. The feces sample included a single swab inserted in the anal sphincter and kept in a 15 mL sterile tube containing 3 mL of VTM. After sampling, all VTM tubes were transferred to the laboratory on the same day where the VTM was aliquoted and stored at –70˚C. Two follow-up visits were done at 15 (± 3 days) and 30 (± 7 days) after the first visit. During the following visits, dogs and cats were submitted to the same clinical examination and collection of biological samples.

Serum samples from cats were submitted to the following tests: detection of antibodies to feline immunodeficiency virus (FIV), and detection of antigen of feline leukemia virus (FeLV) antigens by an enzyme immunoassay using the (Alere FIV Ac/FeLV Ag Test Kit), according to manufacturer instructions.

Serum samples of dogs were submitted to the following tests: rapid test for detection of ehrlichiosis, heartworm disease, Lyme disease, and anaplasmosis (4Dx Plus®—IDEXX Laboratories) according to manufacturer instructions.

**Detection of SARS-CoV-2 in animal samples.** The RNA extraction of VTM samples from NP/OP and rectal swabs was performed as described for human samples above. Then, the RNA samples were tested individually for SARS-CoV-2 using the same SARS-CoV-2 detection kit as described for human samples as well. However, different from human samples, pet samples were also tested by a RT-PCR designed to amplify two targets of gene N of SARS-CoV-2, as previously described [23]. Pet samples were only considered positive for SARS-CoV-2 when at least two genes were amplified. All samples with only one target amplified were considered inconclusive.

Following, all inconclusive and positive samples were submitted to further analyses for virus infection confirmation with attempts to sequence the virus. To achieve this, samples were submitted first to One-Step RT-PCR Enzyme MixKit (Qiagen) using set six pairs of primers targeting fragments of ORF of SARS-CoV-2 with the total expected size of 1200 base pairs (bp) (S1 Table). All reactions were conducted in Verit Thermo Cycler (Applied Biosystems) with the following conditions: reverse transcription (50˚C, 30 min), reverse transcriptase inactivation, and DNA polymerase activation (95˚C, 15 min), followed by 40 cycles of DNA denaturation (94˚C, 40 s) and annealing (55˚C, 40 s) and extension (72˚C, 1 min) and one cycle of the final extension step (72˚C, 1 min). Following, the second PCR using Phusion RT-PCR Enzyme Mixkit (Sigma-Aldrich), the same pair of primers and 1 uL of the amplified product

as a template were used in the following conditions: denaturation (98˚C, 30 s), followed by 35 cycles of DNA denaturation (98˚C, 15 s) and annealing (55˚C, 15 s), extension (72˚C, 30 s) and one cycle of the final extension step (72˚C, 5 min).

Amplicons (~440 bp) were visualized on 1.5% agarose gels stained with SYBR™ Safe DNA Gel Stain (Thermo Fisher Scientific). Subsequently, PCR products of the expected size were purified with ExoSAP-IT™ PCR Product Cleanup Reagent or with QIAquick Gel Extraction kit (Qiagen).

Purified products were subjected to Sanger sequencing reactions using the BigDye™ Terminator v3.1 Cycle Sequencing Kit (Applied Biosystems), according to manufacturer specifications and specific primers. The readings were performed by the 96-capillary 3730 xl DNA Analyzer® (Applied Biosystems) according to the protocols developed by Otto et al. [24]. Obtained sequences were analyzed and aligned with the complete reference sequence of hCoV-19/Wuhan/WIV04/2019 (EPI_ISL_402124) in the Sequencher® 1.5. Software. The final consensus sequences were deposited in the GISAID database (https://www.gisaid.org/).

**Plaque Reduction Neutralization Test (PRNT$_{90}$).** The PRNT is a highly specific serologic test for detection of neutralizing antibodies to SARS-CoV-2 [25]. All serum and plasma samples were heat-inactivated at 56˚C for 30 minutes to inactivate the complement system and then tested by the 90% plaque reduction neutralization test (PRNT$_{90}$) for SARS-CoV-2. Infectious SARS-CoV-2 used for PRNT$_{90}$ was isolated from a human patient from Rio de Janeiro (EPI_ISL_414045) and provided by the LVRS, Fiocruz. According to the World Health Organization (WHO) laboratory biosafety guidance [26], all manipulation of infectious SARS-CoV-2 was performed in a multi-user research facility of biosafety level 3 platform of Oswaldo Cruz Institute (IOC)/Fiocruz. Briefly, the samples were initially screened at a dilution of 1:10 and those that neutralized virus challenge by at least 90% in Vero CCL-81 cells were further tested at serial two-fold dilutions that ranged from 1:10–1:320 to determine 90% end-point titers. Samples were considered seropositive when a dilution of at least 1:20 reduced at least 90% of the formation of viral plaques of SARS-CoV-2, as previously described [8, 27, 28].

**Definition for a confirmed case of animal SARS-CoV-2.** We adopted The World Organization for Animal Health (OIE) definition criteria for confirmed cases (infection) of SARS-CoV-2 in animals. This criteria states viral isolation of SARS-CoV-2 from a sample taken directly from an animal, or identification of SARS-CoV-2 viral RNA in a sample taken directly from an animal that either target at least two specific genomic regions at a level indicating the presence of an infectious virus, or targets a single genomic region followed by sequencing of a secondary target [29]. Animals with only positive PRNT$_{90}$ were considered as seropositive for SARS-CoV-2.

**Statistical analysis.** Descriptive analysis of the study participants and their pets was performed. Frequencies and percentages were reported for categorical variables. The chi-square test or Fisher's exact test were used for categorical variables, and the Student t-test or the Mann Whitney U test for nonparametric variables. Open-ended questions were listed and coded for meaningful comparisons of their distribution. Univariable and multivariable logistic regressions were conducted to assess the Odds Ratio (OR) that a risk factor was associated with animal SARS-CoV-2 infection. Covariates that were significant at p-value < 0.10 in bivariable models were selected to be adjusted in a multivariable model. Variables significant at the 5% level were kept in the final model. Statistical analysis was conducted using statistical software R, version 3.6 (R Core Team, 2020) and IBM SPSS Statistics 22.0.

**Ethics.** The project was approved by the Brazilian National Commission of Ethics in Research (CONEP) under the number CAAE: 30648220.7.0000.5262, and by the Ethics Commission on the Use of Animals (CEUA/Fiocruz), License LW-6/2020.

## Results

### Characteristics of the human participants and their companion pets

Between May 2nd and October 7th, 2020, 102 (42 men, 60 women) patients that tested positive by RT-PCR for SARS-CoV-2 were assessed for eligibility (S1 Fig). Within this group, 21 index (8 men, 13 women) participants from 21 households with their 39 companion pets (29 dogs and 10 cats) were enrolled in the study (S1 Fig). There was not a statistically significant difference in age and sex among enrolled and not enrolled participants. Two human index cases, two dogs, and three cats were lost to follow up.

The demographic and clinical characteristics of the human component of the study are shown in S2 Table. The cohort consists mainly of self-reported women (n = 13, 61.9%), single (n = 10, 47.6%), with a median age of 39.9 years (IQR; 32.7–48.9), and with white race/ethnicity (n = 14, 66.6%). Most individuals reported university/post-graduation levels of education (n = 15, 71.4%). All but one index case required hospitalization. The median number of residents per household was 4 (IQR; 3–4.5), with a median of 2 patients (IQR; 1–2) with a confirmed diagnosis of COVID-19. The median cycle threshold (Ct) at COVID-19 diagnosis was 30.67 (IQR; 19.77–36.08). All participants lived in the metropolitan region of Rio de Janeiro.

Among the 39 animals enrolled in the study (n = 10 cats, 25.6% and n = 29 dogs, 74.4%), similar distribution between sex (n = 19 males, 48.7% and n = 20 females, 51.3%) and breed (n = 19, 48.7% with breed and n = 20, 51.3% mongrel) were observed. The median age at enrollment was 5 years (IQR; 2.0–8.5).

None of the cases of human patients diagnosed with COVID-19 needed hospitalization and the infection resolved itself with no signs of sequelae. Also, there was no need for specific medical care during the follow-up appointments of the animals positive for SARS-CoV-2. No human or animal deaths were observed during the study.

### Detection of SARS-CoV-2 infection among dogs and cats

Median days between the onset of symptoms of the index human case to the first animal sample collection was 16 days (IQR; 12–20), with a minimum and maximum of three and 31 days, respectively. The median days from the onset of symptoms of the related index human case to the day of the third sample collection was 43 days (IQR; 40–51) with a minimum and maximum of 31 and 64 days, respectively. The nasopharyngeal/oropharyngeal (NP/OP) and rectal swabs were collected from 39 animals. However, not all animals were available for all three planned samplings (S3 Table). A total of 212 samples was submitted individually to real-time RT-PCR for SARS-CoV-2.

One target gene was amplified in 12 samples (5.7%) and these samples were considered inconclusive (Table 1). Two and three target genes were amplified in three (1.4%) and eight (3.8%) samples, respectively, from six different animals (cats: 17044 and 17189, dogs: 17109, 17110, 17111, 17172), and were considered positive (Table 1). Median Ct values were: 37.52 (IQR; 36.08–38.42, min 33.92, max 43.23).

Our further efforts to amplify and obtain sequences of short segments of the SARS-CoV-2 virus resulted in the confirmation of nine previously inconclusive and five real-time RT-PCR-positive samples. Using the nucleotide sequencing of short fragments approach, we were able to confirm positive six additional animals (cats: 17032 and 17036, and dogs: 17037, 17050, 17064, 17079) that were previously considered inconclusive by real-time RT-PCR. It is noteworthy, that not all real-time RT-PCR-positive samples were successfully sequenced (Table 1). A total of eight (28%) dogs and four (40%) cats from 10 (47.6%) households were positive for SARS-CoV-2 confirmed by real-time RT-PCR and/or sequencing (Table 1).

**Table 1. Dogs and cats that tested positive for SARS-CoV-2 by real-time RT-PCR and/or sequencing by Sanger method and tested seropositive for antibodies by plaque reduction neutralization test, between May 2nd, 2020 and October 7th, 2020 (metropolitan region of the state of Rio de Janeiro, Brazil).**

| Household ID (Animal ID) | Species | Sample[a] | Type of sample[b] | Sample Date | Days from the human onset of symptoms | Ct values E | Ct values N1 | Ct values N2 | Sequenced by Sanger | PRNT Test titers Sample 1 | PRNT Test titers Sample 2 | PRNT Test titers Sample 3 |
|---|---|---|---|---|---|---|---|---|---|---|---|---|
| 5 (17032) | Cat | S1 | NP/OP | 21-May-2020 | 27 | 35.49 | nd | nd | Confirmed | <10 | no sample | no sample |
| | | S1 | R | 21-May-2020 | 27 | 37.49 | nd | nd | Negative | | | |
| 8 (17036) | Cat | S1 | NP/OP | 28-May-2020 | 18 | 38.48 | nd | nd | Confirmed | <10 | No sample | No sample |
| | | S1 | R | 28-May-2020 | 18 | nd | nd | nd | | | | |
| 8 (17037) | Dog | S1 | NP/OP | 28-May-2020 | 18 | 38.45 | nd | nd | Confirmed | <10 | <10 | <10 |
| | | S1 | R | 28-May-2020 | 18 | nd | nd | nd | | | | |
| | | S2 | NP/OP | 10-Jun-2020 | 31 | nd | nd | nd | | | | |
| | | S2 | R | 10-Jun-2020 | 31 | nd | nd | nd | | | | |
| | | S3 | NP/OP | 24-Jun-2020 | 45 | nd | nd | nd | | | | |
| | | S3 | R | 24-Jun-2020 | 45 | nd | nd | nd | | | | |
| 10 (17043) | Dog | S1 | NP/OP | 05-Jun-2020 | 31 | nd | nd | nd | | 20 | No sample | No sample |
| | | S1 | R | 05-Jun-2020 | 31 | nd | nd | nd | | | | |
| | | S2 | NP/OP | 19-Jun-2020 | 45 | nd | nd | nd | | | | |
| | | S2 | R | 19-Jun-2020 | 45 | nd | nd | nd | | | | |
| | | S3 | NP/OP | 03-Jul-2020 | 59 | nd | nd | nd | | | | |
| | | S3 | R | 03-Jul-2020 | 59 | nd | nd | nd | | | | |
| 10 (17044) | Cat | S1 | NP/OP | 05-Jun-2020 | 31 | nd | nd | nd | | 80 | 80 | No sample |
| | | S1 | R | 05-Jun-2020 | 31 | nd | nd | nd | | | | |
| | | S2 | NP/OP | 19-Jun-2020 | 45 | 38.74 | 36.56 | 35.65 | Negative | | | |
| | | S2 | R | 19-Jun-2020 | 45 | nd | nd | nd | | | | |
| | | S3 | NP/OP | 03-Jul-2020 | 59 | nd | nd | nd | | | | |
| | | S3 | R | 03-Jul-2020 | 59 | nd | nd | nd | | | | |
| 11 (17050) | Dog | S1 | NP/OP | 04-Jun-2020 | 14 | nd | nd | nd | | <10 | <10 | <10 |
| | | S1 | R | 04-Jun-2020 | 14 | nd | nd | nd | | | | |
| | | S2 | NP/OP | 17-Jun-2020 | 27 | nd | nd | nd | | | | |

(*Continued*)

**Table 1.** (Continued)

| Household ID (Animal ID) | Species | Sample[a] | Type of sample[b] | Sample Date | Days from the human onset of symptoms | Ct values E | Ct values N1 | Ct values N2 | Sequenced by Sanger | PRNT Test titers Sample 1 | PRNT Test titers Sample 2 | PRNT Test titers Sample 3 |
|---|---|---|---|---|---|---|---|---|---|---|---|---|
| | | S2 | R | 17-Jun-2020 | 27 | 37.41 | nd | nd | Negative | | | |
| | | S3 | NP/OP | 02-Jul-2020 | 42 | 42.61 | nd | nd | Confirmed | | | |
| | | S3 | R | 02-Jul-2020 | 42 | nd | nd | nd | | | | |
| 20 (17064) | Dog | S1 | NP/OP | 30-Jun-2020 | 16 | nd | nd | nd | | <10 | <10 | <10 |
| | | S1 | R | 30-Jun-2020 | 16 | nd | nd | nd | | | | |
| | | S2 | NP/OP | 15-Jul-2020 | 31 | 43.23 | nd | nd | Confirmed | | | |
| | | S2 | R | 15-Jul-2020 | 31 | 38.43 | nd | nd | Confirmed | | | |
| | | S3 | NP/OP | 27-Jul-2020 | 43 | nd | nd | nd | | | | |
| | | S3 | R | 27-Jul-2020 | 43 | nd | nd | nd | | | | |
| 32 (17079) | Dog | S1 | NP/OP | 20-Jul-2020 | 11 | 38.43 | nd | nd | Confirmed | <10 | <10 | <10 |
| | | S1 | R | 20-Jul-2020 | 11 | nd | nd | nd | | | | |
| | | S2 | NP/OP | 05-Aug-2020 | 27 | nd | nd | nd | | | | |
| | | S2 | R | 05-Aug-2020 | 27 | nd | nd | nd | | | | |
| | | S3 | NP/OP | 20-Aug-2020 | 42 | 37.24 | nd | nd | Confirmed | | | |
| | | S3 | R | 20-Aug-2020 | 42 | nd | nd | nd | | | | |
| 71 (17109) | Dog | S1 | NP/OP | 01-Sep-2020 | 21 | nd | nd | nd | | <10 | <10 | <10 |
| | | S1 | R | 01-Sep-2020 | 21 | nd | nd | nd | | | | |
| | | S2 | NP/OP | 17-Sep-2020 | 37 | nd | nd | nd | | | | |
| | | S2 | R | 17-Sep-2020 | 37 | nd | nd | nd | | | | |
| | | S3 | NP/OP | 01-Oct-2020 | 51 | 38.01 | 37.54 | nd | Confirmed | | | |
| | | S3 | R | 01-Oct-2020 | 51 | 38.46 | 37.38 | 38.14 | Confirmed | | | |
| 71 (17110) | Dog | S1 | NP/OP | 01-Sep-2020 | 21 | 41.47 | 37.75 | 38.73 | Negative | <10 | <10 | <10 |
| | | S1 | R | 01-Sep-2020 | 21 | nd | nd | nd | | | | |
| | | S2 | NP/OP | 17-Sep-2020 | 37 | nd | 37.59 | nd | Negative | | | |
| | | S2 | R | 17-Sep-2020 | 37 | nd | nd | nd | | | | |
| | | S3 | NP/OP | 01-Oct-2020 | 51 | 37.88 | 38.41 | 38.05 | Negative | | | |

(*Continued*)

**Table 1.** (Continued)

| Household ID (Animal ID) | Species | Sample[a] | Type of sample[b] | Sample Date | Days from the human onset of symptoms | Ct values E | Ct values N1 | Ct values N2 | Sequenced by Sanger | PRNT Test titers Sample 1 | PRNT Test titers Sample 2 | PRNT Test titers Sample 3 |
|---|---|---|---|---|---|---|---|---|---|---|---|---|
| | | S3 | R | 01-Oct-2020 | 51 | 36.97 | nd | nd | Confirmed | | | |
| 76 (17111) | Dog | S1 | NP/OP | 02-Sep-2020 | 14 | nd | nd | nd | | <10 | <10 | No sample |
| | | S1 | R | 02-Sep-2020 | 14 | nd | nd | nd | | | | |
| | | S2 | NP/OP | 15-Sep-2020 | 27 | nd | nd | nd | | | | |
| | | S2 | R | 15-Sep-2020 | 27 | nd | nd | nd | | | | |
| | | S3 | NP/OP | 01-Oct-2020 | 43 | 37.90 | 36.47 | 37.61 | Negative | | | |
| | | S3 | R | 01-Oct-2020 | 43 | nd | 35.64 | 35.05 | Confirmed | | | |
| 92 (17172) | Dog | S1 | NP/OP | 02-Oct-2020 | 20 | 35.90 | 36.01 | 35.94 | Confirmed | <10 | <10 | <10 |
| | | S1 | R | 02-Oct-2020 | 20 | nd | nd | nd | | | | |
| | | S2 | NP/OP | 15-Oct-2020 | 33 | nd | 36.29 | 36.11 | Negative | | | |
| | | S2 | R | 15-Oct-2020 | 33 | nd | nd | nd | | | | |
| | | S3 | NP/OP | 06-Nov-2020 | 55 | nd | nd | nd | | | | |
| | | S3 | R | 06-Nov-2020 | 55 | nd | nd | nd | | | | |
| 82 (17189) | Cat | S1 | NP/OP | 20-Oct-2020 | 20 | 33.92 | 36.55 | 35.10 | Confirmed | 80 | No sample | No sample |
| | | S1 | R | 20-Oct-2020 | 20 | 37.67 | 36.10 | 35.65 | Negative | | | |

ID: Identification number.

[a] S1—Swab from the first collection, S2 –Swabs from the second collection; S3 –Swabs from the third collection.

[b] NP/OP–Nasopharyngeal/Oropharyngeal swab; R–Rectal swab; E: envelope; N1: nucleocapsid 1; N2: nucleocapsid 2; RdRP: RNA dependent RNA polymerase; PRNT: Plaque Reduction Neutralization Test; nd–not detected.

Regarding the persistence of SARS-CoV-2 RNA in dogs and cats, we have found that three out of eight (37.5%) dogs had SARS-CoV-2 RNA detected in NP/OP samples for 14 to 31 days after the first positive sample. One of these dogs had tested positive for both N1 and N2 targets, but the other two were positive only for one of the two targets. Dogs were positive for SARS-CoV-2-RNA from 11 to 51 days after the beginning of human index COVID-19 case symptoms. Regarding the persistence of SARS-CoV-2 RNA in cats, none of them had RNA persistence detected (Tables 1 and 2).

## Plaque reduction neutralization tests

To investigate SARS-CoV-2 exposure, 100 samples including serum (n = 92) and plasma (n = 8) of 39 pets were tested by $PRNT_{90}$ for the detection of SARS-CoV-2-specific neutralizing antibodies. From these, four (4%) samples presented $PRNT_{90}$ titer $\geq 10$, including 20 (n = 1), 80 (n = 3) from three different animals, which includes one dog and two cats. These two cats

**Table 2. Fragment size (nucleotides) and position according to the reference strain SARS-Cov-2 (19/Wuhan/WIV04/2019 (WIV04) (EPI_ISL_402124)) of virus obtained from animals in the present study.**

| Household Number (Animal ID) | Species | Sample[a] | Type of sample[b] | ORF1b Start | ORF1ab End | ORF1ab Size (nt) | Genom cover | ORF3ab Start | ORF3ab End | ORF3ab Size (nt) | Genom cover | Name | GISAID Accession number |
|---|---|---|---|---|---|---|---|---|---|---|---|---|---|
| 5 (17032) | Cat | S1 | NP/OP | | | | | 24908 | 25312 | 405 | partial | hCoV-19/cat/Brazil/RJ-A002N/2020 | EPI_ISL_848070 |
| 8 (17036) | Cat | S1 | NP/OP | | | | | 25518 | 25860 | 343 | partial | hCoV-19/cat/Brazil/RJ-A003N/2020 | EPI_ISL_848071 |
| 8 (17037) | Dog | S1 | NP/OP | 15087 | 15772 | 686 | partial | | | | | hCoV-19/dog/Brazil/RJ-A004N/2020 | EPI_ISL_848072 |
| 11 (17050) | Dog | S3 | NP/OP | 15087 | 15439 | 352 | partial | 24898 | 25313 | 416 | | hCoV-19/dog/Brazil/RJ-A016NS3/2020 | EPI_ISL_848076 |
| 20 (17064) | Dog | S2 | NP/OP | 15095 | 15759 | 665 | partial | 25530 | 25912 | 383 | | hCoV-19/dog/Brazil/RJ-A118/2020 | EPI_ISL_848073 |
| | | S2 | R | | | | | 24907 | 25316 | 410 | partial | hCoV-19/dog/Brazil/RJ-A119/2020 | EPI_ISL_848074 |
| 32 (17079) | Dog | S1 | NP/OP | 15087 | 15451 | 365 | partial | | | | | hCoV-19/dog/Brazil/RJ-A136/2020 | EPI_ISL_848075 |
| | | S3 | NP/OP | 15360 | 16029 | 670 | partial | 25530 | 25924 | 395 | partial | hCoV-19/dog/Brazil/RJ-A301/2020 | EPI_ISL_848077 |
| 71 (17109) | Dog | S3 | NP/OP | | | | | 24905 | 25310 | 406 | partial | hCoV-19/dog/Brazil/RJ-A398/2020 | EPI_ISL_848078 |
| | | S3 | R | | | | | 24904 | 25309 | 406 | partial | hCoV-19/dog/Brazil/RJ-A399/2020 | EPI_ISL_848079 |
| 71 (17110) | Dog | S3 | R | | | | | 24908 | 25309 | 402 | partial | hCoV-19/dog/Brazil/RJ-A401/2020 | EPI_ISL_848080 |
| 76 (17111) | Dog | S3 | R | 15661 | 16026 | 366 | partial | | | | | hCoV-19/dog/Brazil/RJ-A403/2020 | EPI_ISL_848081 |
| 92 (17172) | Dog | S1 | NP/OP | 15661 | 15915 | 255 | partial | | | | | hCoV-19/dog/Brazil/RJ-A404/2020 | EPI_ISL_848082 |
| 82 (17189) | Cat | S1 | NP/OP | 15770 | 16016 | 247 | partial | | | | | hCoV-19/cat/Brazil/RJ-A408/2020 | EPI_ISL_848083 |

ID: Identification number.

[a] S1—Swab from the first collection, S2 –Swabs from the second collection; S3 –Swabs from the third collection.

[b] NP/OP–Nasopharyngeal/Oropharyngeal swab; R–Rectal swab; ORF1ab: Open Reading Frame 1ab; ORF3ab: Open Reading Frame 3ab; nt: Nucleotide. hCoV-19/Wuhan/WIV04/2019 (WIV04) (EPI_ISL_402124).

also had SARS-CoV-2 detection in an NP/OP sample (animals 17044 and 17189, Table 1). Neutralizing antibodies for SARS-CoV-2 were detected in cats after 20 and 31 days of human COVID-19 index case onset of symptoms. The only seropositive dog presented neutralizing antibodies 31 days after the onset of symptoms of its related human COVID-19 case (Table 1). Three dogs presented $PRNT_{90}$ titer 10 and according to the criterion of seropositivity were considered inconclusive.

## Prevalence of past or current dogs' vector-borne and viral felines' infections

Of the 29 dogs tested, eight (27.6%) were reactive. From these, specific heartworm antigen was detected in one animal (3.5%), and antibodies for *Ehrlichia canis/ewingii* were detected in five (17.2%). Among the dogs that were seropositive for *Ehrlichia* spp., only one animal had a blood count suggestive of acute ehrlichiosis for presenting thrombocytopenia and anemia. Three (10.3%) animals were reactive for *Anaplasma phagocytophilum/platys* antibodies, and one of them (animal 17064, Table 3) also tested positive for SARS-CoV-2 by molecular methods. All dogs tested negative for *Borrelia burgdorferi* antibodies. Among cats, all animals tested negative for FIV antibodies and FeLV antigen by enzyme immunoassays.

## Animal clinical signs and laboratory alterations

Among the 13 animals infected/seropositive, six (46.2%) had clinical signs that included sneezing, coughing, diarrhea, nasal discharge, regional lymphadenopathy, external otitis, perianal mucosa inflammation, and hyperemic spots on the tongue (Table 3). Also, hematological, liver and renal function tests were similar among the 39 animals. Laboratory abnormalities of the 13 animals infected/seropositive are shown in Table 3.

## Factors associated with animal SARS-CoV-2 infection

Animal SARS-CoV-2 infection was associated with neutering (OR = 12.37; 95%CI 3.28–63.92), cleaning the animal after walking (OR = 14.0; 95% CI 2.10–177.56), sharing the bed with the index human COVID-19 case (OR = 8.80; 95% CI 2.53–34.69), and if the pet spent most of the time indoors (OR = 6.35; 95%CI 1.30–67.16) (Table 4).

In the final multiple regression model, the variables which remained as associated factors to animal SARS-CoV-2 infection were neutering (adjusted odds ratio [aOR] 22.68; 95% CI 4.20–263.93) and sharing the bed with the index human COVID-19 case (aOR 17.17; 95% CI 3.23–188.62) (Table 4).

## Discussion

By combining two extremely sensitive molecular methods (the real-time RT-PCR and Sanger sequencing) to a highly specific serological assay ($PRNT_{90}$), we have demonstrated that dogs and cats living in the same household as their owners with COVID-19 can be exposed and infected by SARS-CoV-2. The persistence of SARS-CoV-2 RNA could be evidenced in some animals. Also, unspecified clinical signs (when presented) in animals were mild and reversible, with mainly respiratory and gastrointestinal manifestations. Data presented here suggest that close contact with human COVID-19 cases is a major risk factor for SARS-CoV-2 infection in companion animals.

The frequency of positive cats (40%), dogs (28%), and households (47.6%) was higher than similar studies using RT-PCR and/or sequencing around the world [8, 9, 11, 15, 17, 30]. In these studies, the frequencies of SARS-CoV-2 infection confirmed by molecular methods

**Table 3. Clinical-epidemiological characteristics and laboratory alterations of 13 animals infected with or seropositive for SARS-CoV-2 between May 2nd, 2020 and October 7th, 2020 (metropolitan region of the state of Rio de Janeiro, Brazil).**

| Household ID (Animal ID) | Species | Breed | Sex | Age (years) | Castration | Vaccination (CCoV or FCoV) | Number of Household pets | Coinfection[a] | Clinical signs (visit) | Laboratory alterations (visit) |
|---|---|---|---|---|---|---|---|---|---|---|
| 5 (17032) | Cat | Mongrel | M | 3 | Yes | No | 1 (1 cat) | No | Sneeze and Lymphadenomegaly (1) | ↓Platelets (1) |
| 8 (17036) | Cat | Mongrel | F | 1 | Yes | No | 3 (2 cats and 1 dog) | No | None | None |
| 8 (17037) | Dog | Mongrel | F | 8 | Yes | CCoV | 3 (2 cats and 1 dog) | No | None | ↑Albumin (1,2,3), ↑A/G ratio (1,3), ↑Total Protein (2) |
| 10 (17043) | Dog | Poodle | F | 16 | Yes | CCoV | 7 (5 cats and 2 dogs) | No | Sneezing, coughing, nasal discharge, diarrhea, seborrheic dermatitis (1). Purulent nasal discharge and urinary tract infection with hematuria (2). Nasal discharge (3). | ↓Platelets (2), ↑Leukocytes (3), ↑BUN (1,2,3) |
| 10 (17044) | Cat | Mongrel | M | 2 | No | No | 7 (5 cats and 2 dogs) | No | Gingivitis (1). Gingivitis, mandibular lymphadenomegaly (2). Mandibular lymphadenomegaly and severe periodontitis (3) | ↓Platelets (1,2) |
| 11 (17050) | Dog | Mongrel | F | 1 | No | CCoV | 2 dogs | No | None | ↓Platelets (1,2,3), anemia (2), ↓Leukocytes (2) ↑A/G ratio (1,2,3), ↓Globulin (1,2,3), ↓Total Protein (3) |
| 20 (17064) | Dog | Yorkshire terrier | F | 11 | Yes | CCoV | 2 dogs | Anaplasma | External otitis (1) | ↑A/G ratio (1), ↓Globulin (1), ↑Total Protein (2), ↑Albumin (2) |
| 32 (17079) | Dog | Shih-tzu | M | 4 | No | CCoV | 1 dog | No | None | ↑AST (2), ↑ A/G ratio (2,3) |
| 71(17109) | Dog | Mongrel | M | 8 | Yes | CCoV | 2 dogs | No | Perianal Mucosa Inflammation (2,3) | ↓Platelets (1,2), ↑AP (1,2), ↓Total Protein (2), ↓Globulin (2), ↑Total Protein (3), ↑Albumin (3) |
| 71 (17110) | Dog | Mongrel | F | 5 | Yes | CCoV | 2 dogs | No | Hyperpigmentation, Hyperkeratosis, and thickening of the perianal region (3) | ↓Platelets (1,2) |
| 76 (17111) | Dog | Pinscher | M | 5 | Yes | CCoV | 1 dog | No | None | ↓Platelets (1,2), ↑Total Protein (1,2), ↑Albumin (1,2), ↑AST (2), ↑Globulin (2) |
| 92 (17172) | Dog | Labrador | F | 15 | Yes | CCoV | 1 dog | No | None | ↓Platelets (1,2,3), ↑ALT (1,2,3) |
| 82 (17189) | Cat | Mongrel | M | 4 | Yes | No | 1 cat | No | Hyperemic spots on the tongue (1) | ↓Platelets (1), ↑ALT (1), ↑BUN (1), ↑Creatinine (1) |

M: Male, F: Female, CCoV: Canine coronavírus, FCoV: Feline coronavírus, A/G ratio: Albumin-Globulin (A/G) Ratio.

[a] Rapid test for detection of ehrlichiosis, heartworm disease, Lyme disease, and anaplasmosis (4Dx Plus®—IDEXX Laboratories) and antibodies to feline immunodeficiency virus (FIV), and feline leukemia virus (FeLV) antigens by enzyme immunoassay using the Alere FIV Ac/FeLV Ag Test Kit, ALT: alanine aminotransferase, AST: aspartate aminotransferase, AP: alkaline phosphatase, BUN: blood urea nitrogen, ↑: increased, ↓: decreased.

**Table 4. Unadjusted and adjusted odds ratios for factors associated with animal SARS-CoV-2 infection, between May 2nd, 2020 and October 7th, 2020 (metropolitan region of the state of Rio de Janeiro, Brazil), (n = 39).**

| Characteristics | With SARS-CoV-2 infection | | Without SARS-CoV-2 infection | | Crude OR | 95% CI | P value | Adjusted OR | 95% CI | P value |
|---|---|---|---|---|---|---|---|---|---|---|
| | n | % | n | % | | | | | | |
| **Breed** | | | | | | | | | | |
| Mongrel | 8 | 61.5 | 12 | 46.2 | 1 | | | | | |
| With breed | 5 | 38.5 | 14 | 53.8 | 1.87 | 0.61–6.05 | 0.368 | | | |
| **Species** | | | | | | | | | | |
| Canine | 9 | 69.2 | 20 | 76.9 | 1 | | | | | |
| Feline | 4 | 30.8 | 6 | 23.1 | 1.48 | 0.41–5.16 | 0.605 | | | |
| **Sex** | | | | | | | | | | |
| Male | 6 | 46.2 | 13 | 50.0 | 1 | | | | | |
| Female | 7 | 53.8 | 13 | 50.0 | 1.17 | 0.38–3.64 | 0.821 | | | |
| **Animal Age, median (IQR)[a]** | 6.0 (2.5–9.7) | | 4.1 (1.3–8.2) | | 1.09 (0.95–1.25) | | 0.303 | | | |
| **Castration** | | | | | | | | | | |
| No | 2 | 15.4 | 18 | 69.2 | 1 | | | 1 | | |
| **Yes** | **11** | **84.6** | **8** | **30.8** | **12.37** | **3.28–63.92** | **0.004** | **22.68** | **4.20–263.93** | **0.009** |
| **Another feline or canine at home** | | | | | | | | | | |
| No | 4 | 30.8 | 5 | 19.2 | 1 | | | | | |
| Yes | 9 | 69.2 | 21 | 80.8 | 1.87 | 0.50–6.80 | 0.424 | | | |
| **Coinfection[b]** | | | | | | | | | | |
| No | 12 | 92.3 | 19 | 73.1 | 1 | | | | | |
| Yes | 1 | 7.7 | 7 | 26.9 | 4.42 | 0.88–47.22 | 0.189 | | | |
| **Does the animal leave the house for a walk (dogs) or leaves the street unaccompanied by the owners (cats)?** | | | | | | | | | | |
| No | 6 | 46.2 | 15 | 57.7 | 1 | | | | | |
| Yes | 7 | 53.8 | 11 | 42.3 | 1.59 | 0.52–5.00 | 0.497 | | | |
| **Does the animal go to a pet shop for a bath and grooming?** | | | | | | | | | | |
| No | 10 | 76.9 | 15 | 57.7 | 1 | | | | | |
| Yes | 3 | 23.1 | 11 | 42.3 | 2.44 | 0.73–9.58 | 0.245 | | | |
| **Is the animal cleaned after walking on the street?** | | | | | | | | | | |
| No | 1 | 14.3 | 7 | 70.0 | 1 | | | | | |
| **Yes** | **6** | **85.7** | **3** | **30.0** | **14.0** | **2.10–177.56** | **0.039** | | | |
| **Place where the animal spends most of its time** | | | | | | | | | | |
| Outdoors (terrace, backyard) | 1 | 7.7 | 9 | 34.6 | 1 | | | | | |
| **Indoors** | **12** | **92.3** | **17** | **65.4** | **6.35** | **1.30–67.16** | **0.099** | | | |
| **Does the person with the COVID-19 diagnosis share the bed with the animal?** | | | | | | | | | | |
| No | 5 | 38.5 | 22 | 84.6 | 1 | | | | | |
| **Yes** | **8** | **61.5** | **4** | **15.4** | **8.80** | **2.53–34.69** | **0.006** | **17.17** | **3.23–188.62** | **0.014** |
| **Does the person with a COVID-19 diagnosis accept to have their face and/or hands licked by the animal?** | | | | | | | | | | |
| No | 5 | 38.5 | 8 | 30.8 | 1 | | | | | |
| Yes | 8 | 61.5 | 18 | 69.2 | 1.41 | 0.43–4.53 | 0.632 | | | |
| **Proportion of people with COVID-19 diagnosis in the household, median (IQR)[a]** | 0.66 (0.25–0.71) | | 0.33 (0.25–0.75) | | 1.66 (0.25–11.14) | | 0.658 | | | |

*(Continued)*

**Table 4.** (Continued)

| Characteristics | With SARS-CoV-2 infection | | Without SARS-CoV-2 infection | | Crude OR | 95% CI | P value | Adjusted OR | 95% CI | P value |
|---|---|---|---|---|---|---|---|---|---|---|
| | n | % | n | % | | | | | | |
| **If a young person under the age of 18 lives in the residence, does he or she was diagnosed with COVID-19?** | | | | | | | | | | |
| No | 3 | 37.5 | 12 | 66.7 | 1 | | | | | |
| Yes | 5 | 62.5 | 6 | 33.3 | 3.33 | 0.80–15.46 | 0.174 | | | |
| **Human cycle threshold from NP/OP swab at the diagnosis, median (IQR)[a]** | 35.8 (21.8–38.5) | | 34.3 (18.8–36.1) | | 1.04 (0.97–1.12) | | 0.413 | | | |
| **Duration of detectable SARS-CoV-2 RT-PCR from human NP/OP swab, median (IQR)[a] (days)** | 27.5 (16.3–41.0) | | 20.0 (18.0–38.5) | | 1.02 (0.96–1.08) | | 0.674 | | | |

95%CI = 95% confidence interval; OR = odds ratio; IQR: Interquartile Range.

[a] Mann-Whitney test.

[b] Rapid test for detection of ehrlichiosis, heartworm disease, Lyme disease and anaplasmosis (4Dx Plus®—IDEXX Laboratories) and antibodies to feline immunodeficiency virus (FIV), and feline leukemia virus (FeLV) antigens by enzyme immunoassay using the Alere FIV Ac/FeLV Ag Test Kit; Human Cycle threshold from NP/OP swab at the diagnosis = 5; Persistence of detectable SARS-CoV-2 RT-PCR from NP/OP swab of the human index case = 16.

were: 17.6% in cats, 1.7% in dogs, and 10.3% of households in Texas, United States of America [17]; 4% in cats and all dogs negative in France [9]; All dogs and cats negative in Italy [30]; 12% in cats and 13% in dogs from Hong Kong, China [8, 15]; and finally, 12% in cats and all dogs negative in La Rioja, Spain [11].

The higher frequency of positivity presented here may be explained by the longitudinal approach with serial sample collections of all animals in a period of nearly 30 days after the first collection. Also, all animals tested were from households with at least one human COVID-19 case and the animal samples were collected close to the human onset of symptoms.

According to our data, 41.7% (5/12) of the animals had detected RT-PCR in the second and third visits. These findings reinforce the importance of longitudinal studies to investigate SARS-CoV-2 infection in pets. The fact that the only study with a similar design [17] found the second-highest frequency of positivity in cats and the third-highest frequency in dogs reinforces this hypothesis. According to a study conducted in Texas, USA, the difference in frequencies may be explained by the different number of animals investigated, as well as by the habits and characteristics of the pets [17]. As reported in Texas, our findings suggest that companion animals from Rio de Janeiro have also been exposed to SARS-CoV-2 in households with at least one human case of COVID-19.

The present study corroborates other similar studies that found a higher positivity in cats when compared to dogs [9, 11, 17, 30]. Together, these results confirm that both dogs and cats are susceptible to SARS-CoV-2 infection, but that cat populations have been more clinically affected, which has been corroborated by experimental studies [31, 32]. The similar susceptibility of both cats and dogs to SARS-CoV-2 may be explained by the fact that both species share with humans the virus receptor Angiotensin-converting enzyme 2 (ACE2). The fact that the ACE2 of cats is more closely related to ACE2 of humans than to the ACE2 of dogs, may explain why cats are more affected by SARS-CoV-2 infection [33].

The persistence of viral RNA in the NP/OP of dogs presented here has already been described in dogs elsewhere [8, 32]. A dog was found positive for SARS-CoV-2 RNA 13 days

after the first positive sample, and up to 28 days after the onset of clinical signs of the related human COVID-19 case in China [8, 32]. In a study conducted in Texas [17], viral RNA persistence was detected only in cats. Two cats were positive by real-time RT-PCR for up to 25 days after the first positive sample, and 32 days after the confirmation of SARS-CoV-2 infection in the owner.

In studies conducted in China and the USA, SARS-CoV-2 RNA persistence was found in cats for 8 to 17 days corresponding to the period of 22 to 34 days after the onset of symptoms of the human COVID-19 case [12, 15]. In the present study, only one of the four infected cats had more than one sample collected and for that reason, we were unable to fully evaluate the RNA persistence. Combining these results, it was possible to demonstrate that the persistence of SARS-CoV-2 RNA in dogs was not rare.

It has been experimentally demonstrated that cats can shed infectious viruses for up to 5–6 days, and that infected cats can horizontally transmit the virus to other cats. On the other hand, the shedding of infectious viruses by dogs has not been reported yet [31, 32]. The molecular evidence presented here is not sufficient to confirm the presence of an infectious virus and may be due to residual fragments of viral RNA from an abortive cycle of replication detected by the highly sensitive molecular methods used here. Therefore, the persistence of viral RNA of up to 32 days in dogs and 25 days in cats should be interpreted with caution and must not be used alone to determine the period of quarantine for dogs and cats infected with SARS-CoV-2. More longitudinal and experimental studies including virus isolation in cell cultures are necessary for a better understanding of the role of cats and dogs in SARS-CoV-2 spreading.

Findings from natural or experimentally infected cats and dogs suggest that these companion animals can produce antibodies for SARS-CoV-2, indicating that they can develop an immunological response against the virus [8–10, 14, 30–32, 34–38]. Compared to our findings, previously reported studies involving the investigation of neutralizing antibodies for SARS-CoV-2 in animals have found higher seroprevalences for SARS-CoV-2 mainly in dogs. The frequencies of seropositivity in dogs were 11.9% (7/59), 12.8% (6/47), and 13.3% (2/15) with titers ranging from 8 to 160 in dogs from the USA [17], Italy [30], and China [15], respectively.

Seroprevalence for neutralizing antibodies for SARS-CoV-2 is usually lower in cats, but it varies greatly from 4% to 43.8% (7/16), with titers ranging from 16 to 2048 [15, 17, 30]. However, it is important to mention that the percentage of neutralization used as criteria of seropositivity in neutralizing antibody studies varies greatly which difficult comparison analysis [8, 15, 17, 30].

Other serological studies using different methodologies such as ELISA, immunoprecipitation test, microsphere immunoassay, neutralization activity measurement, and in-house microneutralization test have detected frequencies of seropositivity for SARS-CoV-2 antibodies ranging from 0% to 23.5% in dogs and cats in France [37], Croatia [36] and China [34]. These different antibody tests used to detect previous exposure of companion animals to SARS-CoV-2 performed all around the world demonstrate that there is a need for the standardization of a serological test to better evaluate and compare serosurveys.

According to our data, the frequencies of dogs and cats seropositive for neutralizing antibodies for SARS-CoV-2 were lower than the frequencies of positivity by RT-PCR (2x inferior in cats and 8x inferior in dogs). Besides, only 16.7% (2/12) of the pets that tested positive by RT-PCR were seropositive for neutralizing antibodies. These results differ from another longitudinal study with a similar design conducted in Texas in which the seroprevalence (18.6%) was higher than the frequency of positivity by RT-PCR (5.2%) [17]. In that study, 50% of these animals that tested positive by RT-PCR were seropositive for neutralizing antibodies [17]. In the same study, the parameter used for sampling dogs and cats was the day when the human

household member was diagnosed as a COVID-19 case. The real-time RT-PCR-positive cases were detected from three to 27 days after human diagnosis and the seropositive animals between five and 93 days after human diagnosis.

In the present study, the parameter used for starting the collection of samples of pets was the onset of symptoms of COVID-19 in the owner. The real-time RT-PCR-positive cases were detected between 18 and 51 days after the onset of symptoms of the human COVID-19 case, and seropositive animals were detected from 18 to 45 days after the onset of symptoms.

Early sampling post-symptom onset in humans is recommended to increase sensitivity for detection of SARS-CoV-2 by RT-PCR and improve the reliability of the result [26]. Also, cats and dogs develop antibodies to SARS-CoV-2 as measured by PRNT as early as seven- and 14-days post-infection. Future studies to define the best time of collection of the samples for real-time RT-PCR and antibody testing for the diagnosis of SARS-CoV-2 infection or exposure in dogs and cats are necessary for a better evaluation of the natural history of SARS-CoV-2 infection in companion animals.

Animals from our study developed unspecified, mild, and reversible signs of disease without important laboratory abnormalities. Respiratory and or gastrointestinal signs were also observed in the cats from Belgium, France, Italy, Spain, and the two cats from the USA [9, 10, 12–14]. The prevalence of past or current vector-borne infections in dogs in our study was almost 30% but was not a risk factor for the animal SARS-CoV-2 infection.

Neutering has been shown to reduce the risk of mammary cancer, pyometra, or testicular cancer and to increase the risk for certain autoimmune disorders [39]. A marked reduction or elimination of sexually dimorphic behaviors of dogs and cats, such as roaming, especially in cats, and decreased aggression (which increases interaction), especially in dogs, are other effects in neutered animals [40–42]. Neutering in the present study was a risk factor for SARS-CoV-2 infection in animals. One hypothesis for this result is that the reduction of roaming leads the pet to stay indoors in closer contact with the owner infected with SARS-CoV-2. Reinforcing this hypothesis, the characteristic of the pet to spent most of the time indoors was also associated with animal SARS-CoV-2 infection in the present study. However, further studies are needed to better clarify the pathophysiology of this association.

Our data suggest that the close contact of the animal with the owner with COVID-19, such as sharing the bed with the owner with COVID-19, seems to be the main risk factor for infection of the animals. Thus, it is recommended that suspected or confirmed humans with COVID-19 avoid close contact with their companion pets and have another member not infected with SARS-CoV-2 of their household to care for them. The use of face masks and basic hygiene measures should be followed when handling and caring for animals.

Our study has some limitations: 1) Even with a longitudinal design including three collections at different times points after the human COVID-19 diagnosis, we could not determine the duration of viral elimination and antibody response in all animals with SARS-CoV-2 infection. Five animals showed evidence of SARS-CoV-2 infection only in the last sample collection, and some animals, especially cats, did not have samples collected in all visits by choice of their owners; 2) The comparison of SARS-CoV-2 nucleotide sequences between human COVID-19 cases and their positive pets was not possible to be evaluated because of the remarkable low viral load detected from all animal samples that tested positive. Because of the low viral load, we were unable to obtain full genome sequences through Next-Generation Sequencing (NGS) (Illumina technology) for better phylogenetic analyses. For that reasons, it must still be elucidated if the animal infection has been caused by the same virus as their tutors as well as whether any viral mutations have been presented in the animal's samples; 3) We were also unable to assess whether the presence of other underlying health problems such as cancer and immunosuppression could be related to the detection of SARS-CoV-2 due to the

good health conditions of the animals enrolled in the study; 4) Data on factors associated with SARS-CoV-2 infection in animals should be interpreted with caution due to the study's small sample size, which can be reflected in the wider confidence intervals observed; 5) Although PRNT using a highly conservative criterion of positivity of 90% neutralization is the most specific test for neutralizing antibodies, without the inclusion of other coronaviruses as differential diagnosis the possibility of cross-reactivity cannot be fully discarded.

The present study highlights the role of the One Health approach in the mitigation and control of the COVID-19 pandemic [43, 44]. The integrated surveillance infrastructure of the Evandro Chagas National Institute of Infectious Diseases and the Regional Reference Laboratory in the Americas for Coronavirus was able to timely monitor and detect the occurrence of SARS-CoV-2 infection in both humans and animals using a multi-professional approach, including veterinarians, physicians, other health professionals, and a statistician The results of our study have been provided to The Brazilian Ministry of Agriculture, Livestock and Food Supply and can contribute to the investigation of animal SARS-CoV-2 hot spots.

In conclusion, this study demonstrated the presence and persistence of SARS-CoV-2 infection in different biological samples of dogs and cats that lived in the same residence of SARS-CoV-2 infected owners. The investigation of the clinical signs associated with the infection and the analysis of the laboratory data of these animals suggests that pets may present from an absence of clinical signs to unspecified and mild transient respiratory and gastrointestinal manifestations, without significantly associated laboratory abnormalities. The results presented here suggest that people diagnosed with COVID-19 should avoid direct contact with their pets for as long as they remain ill and that further longitudinal studies must be carried out to confirm these findings.

## Supporting information

**S1 Fig. Flow diagram of the study.**
(DOCX)

**S1 Table. Primers used to amplify fragments of the genome of SARS-CoV-2 in the present study.**
(DOCX)

**S2 Table. Baseline characteristics of 21 index human study participants between May 2[nd], 2020 and October 7[th], 2020 (metropolitan region of the state of Rio de Janeiro, Brazil).**
(DOCX)

**S3 Table. The number of cats and dogs and samples tested for RT-PCR for the detection of SARS-CoV-2 in the study, between May 2[nd], 2020 and October 7[th], 2020 (metropolitan region of the state of Rio de Janeiro, Brazil).**
(DOCX)

## Acknowledgments

We thank the members of the Multi-user Research Facility of Biosafety Level 3 Platform of Instituto Oswaldo Cruz, Fiocruz, Rio de Janeiro, Brazil, and Genomic Platform—DNA Sequencing—RPT01A (Rede de Plataformas Tecnológicas Fiocruz), Coordenação de Aperfeiçoamento de Pessoal de Nível Superior–Brasil (CAPES), and Dr. Valdiléa Gonçalves Veloso dos Santos, director of the Evandro Chagas National Institute of Infectious Diseases/Fiocruz, who provided institutional support to the study. The authors would also like to thank all animal owners for their consent to participate in the research.

## Author Contributions

**Conceptualization:** Guilherme Amaral Calvet, Sandro Antonio Pereira, Maria Ogrzewalska, Alex Pauvolid-Corrêa, Paola Cristina Resende, Marilda Mendonça Siqueira, Isabella Dib Ferreira Gremião, Rodrigo Caldas Menezes.

**Data curation:** Guilherme Amaral Calvet, Sandro Antonio Pereira, Maria Ogrzewalska, Alex Pauvolid-Corrêa, Paola Cristina Resende, Ana Beatriz Machado Lima, Marilda Mendonça Siqueira, Isabella Dib Ferreira Gremião, Rodrigo Caldas Menezes.

**Formal analysis:** Guilherme Amaral Calvet, Sandro Antonio Pereira, Maria Ogrzewalska, Alex Pauvolid-Corrêa, Paola Cristina Resende, Wagner de Souza Tassinari, Isabella Dib Ferreira Gremião, Rodrigo Caldas Menezes.

**Funding acquisition:** Sandro Antonio Pereira, Marilda Mendonça Siqueira.

**Investigation:** Guilherme Amaral Calvet, Sandro Antonio Pereira, Maria Ogrzewalska, Alex Pauvolid-Corrêa, Paola Cristina Resende, Anielle de Pina Costa, Lucas Oliveira Keidel, Alice Sampaio Barreto da Rocha, Michele Fernanda Borges da Silva, Shanna Araujo dos Santos, Ana Beatriz Machado Lima, Isabella Campos Vargas de Moraes, Artur Augusto Velho Mendes Junior, Thiago das Chagas Souza, Ezequias Batista Martins, Renato Orsini Ornellas, Maria Lopes Corrêa, Isabela Maria da Silva Antonio, Lusiele Guaraldo, Patrícia Brasil, Isabella Dib Ferreira Gremião, Rodrigo Caldas Menezes.

**Methodology:** Guilherme Amaral Calvet, Sandro Antonio Pereira, Maria Ogrzewalska, Alex Pauvolid-Corrêa, Paola Cristina Resende, Wagner de Souza Tassinari, Ana Beatriz Machado Lima, Marilda Mendonça Siqueira, Isabella Dib Ferreira Gremião, Rodrigo Caldas Menezes.

**Project administration:** Guilherme Amaral Calvet, Sandro Antonio Pereira, Maria Ogrzewalska, Isabella Dib Ferreira Gremião, Rodrigo Caldas Menezes.

**Resources:** Guilherme Amaral Calvet, Sandro Antonio Pereira, Anielle de Pina Costa, Michele Fernanda Borges da Silva, Isabella Campos Vargas de Moraes, Ezequias Batista Martins, Lusiele Guaraldo, Patrícia Brasil, Marilda Mendonça Siqueira, Isabella Dib Ferreira Gremião, Rodrigo Caldas Menezes.

**Software:** Guilherme Amaral Calvet, Maria Ogrzewalska, Paola Cristina Resende, Alice Sampaio Barreto da Rocha, Ana Beatriz Machado Lima, Marilda Mendonça Siqueira.

**Supervision:** Guilherme Amaral Calvet, Sandro Antonio Pereira, Maria Ogrzewalska, Fernando do Couto Motta, Marilda Mendonça Siqueira, Isabella Dib Ferreira Gremião, Rodrigo Caldas Menezes.

**Validation:** Guilherme Amaral Calvet, Sandro Antonio Pereira, Maria Ogrzewalska, Alex Pauvolid-Corrêa, Paola Cristina Resende, Ana Beatriz Machado Lima, Fernando do Couto Motta, Marilda Mendonça Siqueira, Isabella Dib Ferreira Gremião, Rodrigo Caldas Menezes.

**Visualization:** Guilherme Amaral Calvet, Sandro Antonio Pereira, Maria Ogrzewalska, Alex Pauvolid-Corrêa, Isabella Dib Ferreira Gremião, Rodrigo Caldas Menezes.

**Writing – original draft:** Guilherme Amaral Calvet, Sandro Antonio Pereira, Maria Ogrzewalska, Alex Pauvolid-Corrêa, Isabella Dib Ferreira Gremião, Rodrigo Caldas Menezes.

**Writing – review & editing:** Guilherme Amaral Calvet, Sandro Antonio Pereira, Maria Ogrzewalska, Alex Pauvolid-Corrêa, Paola Cristina Resende, Wagner de Souza Tassinari, Anielle

de Pina Costa, Lucas Oliveira Keidel, Alice Sampaio Barreto da Rocha, Michele Fernanda Borges da Silva, Shanna Araujo dos Santos, Ana Beatriz Machado Lima, Isabella Campos Vargas de Moraes, Artur Augusto Velho Mendes Junior, Thiago das Chagas Souza, Ezequias Batista Martins, Renato Orsini Ornellas, Maria Lopes Corrêa, Isabela Maria da Silva Antonio, Lusiele Guaraldo, Fernando do Couto Motta, Patrícia Brasil, Marilda Mendonça Siqueira, Isabella Dib Ferreira Gremião, Rodrigo Caldas Menezes.

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
