## [Decision Letter · Decision Letter 0]

10 Mar 2021

PONE-D-21-02395

Investigation of SARS-CoV-2 Infection in Dogs and Cats of Humans Diagnosed with COVID-19 in Rio de Janeiro, Brazil

PLOS ONE

Dear Dr. Calvet,

Thank you for submitting your manuscript to PLOS ONE. After careful consideration, we feel that it has merit but does not fully meet PLOS ONE’s publication criteria as it currently stands. Therefore, we invite you to submit a revised version of the manuscript that addresses the points raised during the review process.

We look forward to receiving your revised manuscript.

Kind regards,

Maged Gomaa Hemida, ph.D

Academic Editor

PLOS ONE

Additional Editor Comments:

Dear Authors,

The reviewers agree that your manuscript have some merits and I really encourage you to address their comments point by point in your revised manuscript if you decided to do so.

Journal Requirements:

Reviewers' comments:

Reviewer's Responses to Questions

**Comments to the Author**

1. Is the manuscript technically sound, and do the data support the conclusions?

Reviewer #1: Yes

Reviewer #2: Yes

Reviewer #3: Yes

2. Has the statistical analysis been performed appropriately and rigorously? 

Reviewer #1: Yes

Reviewer #2: Yes

Reviewer #3: Yes

3. Have the authors made all data underlying the findings in their manuscript fully available?

Reviewer #1: Yes

Reviewer #2: Yes

Reviewer #3: No

4. Is the manuscript presented in an intelligible fashion and written in standard English?

Reviewer #1: Yes

Reviewer #2: Yes

Reviewer #3: Yes

5. Review Comments to the Author

Reviewer #1: The manuscript “Investigation of SARS-CoV-2 Infection in Dogs and Cats of Humans Diagnosed with COVID-19 in Rio de Janeiro, Brazil”. Overall, the manuscript is pertinent and deserves publication after miner revisions (listed below) are done. The manuscript has stated objectives:

1) To assess the exposure, infection, and persistence of SARS-CoV-2 in nasopharyngeal and oropharyngeal secretions (nasal and oral swabs) and feces (rectal swab) of dogs and cats living in households with a COVID-19 human case.

2) To investigate clinical and laboratory alterations (complete blood count and serum biochemistry) in pets associated with the SARS-CoV-2 infection.

3) To evaluate risk factors associated with SARS-CoV-2 infection in pets.

However, the objective 2 does not appear to be pertinent to the context of the manuscript and its results were not really discussed. I recommend the second objective and related parts of the manuscript be removed.

144, 210, 212, 414 etc.: “… human’s samples” should be “… human samples”

145-147: “All VTM samples had viral RNA extracted manually by the QIAamp Viral RNA

Mini kit (Qiagen), or automatedly by Perkin-Elmer Chemagic machine/chemistry…” should be re-phrased because QIAamp Viral RNA kit actually extracts both DNA and RNA, that is total nucleic acids (no DNase treatment). If Perkin-Elmer Chemagic does not include DNase than it would be appropriate to say:

“Nucleic acids were extracted from VTM samples by QIAamp Viral RNA Mini kit (Qiagen), or by Chemagic instrument (Perkin-Elmer)…”

Also please specify the instrument model – was it Chemagic Prime, Chemagic 360 or Chemagic Prepito?

150: “Amplifications were conducted …” should be changed to “Reverse transcription and amplification were conducted…”

213-215: “…tested by a RT-PCR designed to amplify N1 and N2 genes of SARS-CoV-2, as previously described [22]. Pet samples were only considered positive for SARS-CoV-2 when at least two genes were amplified.” This should be re-phrased because: (i) it does not match the reference which describes detection of RdRp, E and N genes by real-time PCR assays; (ii) SARS-CoV-2 does not have N1 and N2 genes.

244-145 “The PRNT is a highly specific serologic test and can be carried out on samples to

confirm the presence of neutralizing antibodies to SARS-CoV-2 [24].” Should be re-phrased because stating “to confirm the presence” implies a screening test was done before, therefore it is more appropriate to state” The PRNT is a highly specific serologic test for detection of neutralizing antibodies to SARS-CoV-2 [24].”

252: research facility of biosafety level 3 platform of IOC/Fiocruz. – define “IOC”

Reviewer #2: This manuscript strengthen previous studies reporting transmission of SARS-CoV-2 from human to dogs and cats, living in the same household . Twenty eight percent of dogs and 40% of cat from 47.6% of households, included in the study, were positive to SARS-CoV-2. The positivity lasted from 11 to 51 days after the human index COVID-19 case onset of symptoms. Risk factors associated with positivity were discussed.

. The longitudinal approach adopted in this study, with the wide sampling window, added more insights on the dynamic of SARS-CoV-2 transmission from human patients to their pets animals. Such finding are of great practical value in protecting pets animals and containing the current of SARS-CoV-2 pandemic, which is in line with one health approach.

Comments to the Authors:

- The material and methods section was extensively written, Please reduce it

- Have you considered sampling dogs and cat from nearby households with negative SARS-CoV-2 owners , as a mean to add a control element to the study.

- Line 58: replace (infected/exposed) with (positive)

- In the entire text replace the word “exposed” and use the world “ infected or positive” because the demarcation between the two words is not sufficiently clear in the current prospective.

-Line 342: the use of plasma in serological testing is not recommended because of the possibility of nonspecific result. Please discard, if any, serological results based on the use of plasma.

Line 514 this sentence should be removed “It was also experimentally demonstrated that cats can become infected and 515 transmit SARS-CoV-2 to other cats” because of its poor relation to the rest of the paragraph.

Reviewer #3: Dear editor,

Many thanks for giving me the opportunity to review the manuscript titled (Investigation of SARS-CoV-2 Infection in Dogs and Cats of Humans Diagnosed with COVID-19 in Rio de Janeiro, Brazil). The manuscript is presenting interesting da about the dynamics of SARS-CoV-2 infection in some companion animals in the close proximity of positive SARS-CoV-2 infected patients. These types of research is of high importance these days to get more information about the roles of these animals in the transmission of SARS-CoV-2 to human as well as the back-zoonotic impacts of human to infect animals. Although the manuscript is presenting large sets of data about the dynamics of SARS-CoV-2 infection in some pets in household of positive cases, some points should be considered before acceptance is granted.

1- One of the major concerns is the lack of description and comparison of the obtained sequences from animals to that of the circulating strains in humans at the time of their longitudinal study. It is highly recommended to sequence the full-length genome or at least the full-length S gene sequences of some isolates from animals and compare it to that of human. Such information will help great impacts in improving the quality of this manuscript. Meanwhile, presenting theses sequences and making them publically available is of high demands.

2- The authors must map in more precise details the sampling schedules from both human and animals.

3- A detailed description of the outcomes of the viral infection in animals in correlation to human must be presented.

4- The statistical analysis in the M&M is long. Please consider shortening this section and use key references instead.

5- The discussion section is very long. Some paragraphs are too vague. Please consider shortening theses sections and mainly focusing this section on discussing your findings in comparison to other international groups that have conduced similar studies especially from USA, Asia, and Europe.

6- Please consider deleting Objectiv-2 since it is not adding too much to the main theme of this manuscript.

7- Please try to highlight the roles of the One Health concept as an important approach for studying this type of research and its potential roles in the mitigation and control of the emerging and re-emerging diseases.

6. PLOS authors have the option to publish the peer review history of their article (what does this mean?). If published, this will include your full peer review and any attached files.

Reviewer #1: No

Reviewer #2: **Yes: **Abdulmohsen A. Alnaeem

Reviewer #3: No

---

## [Author Response · Author response to Decision Letter 0]

15 Mar 2021

If there are ethical or legal restrictions on sharing a de-identified data set, please explain them in detail (e.g., data contain potentially identifying or sensitive patient information) and who has imposed them (e.g., an ethics committee). Please also provide contact information for a data access committee, ethics committee, or other institutional body to which data requests may be sent.

If there are no restrictions, please upload the minimal anonymized data set necessary to replicate your study findings as either Supporting Information files or to a stable, public repository and provide us with the relevant URLs, DOIs, or accession numbers. Please see http://www.bmj.com/content/340/bmj.c181.long for guidelines on how to de-identify and prepare clinical data for publication. For a list of acceptable repositories, please see http://journals.plos.org/plosone/s/data-availability#loc-recommended-repositories.

Yes. As we collected several personally identifiers and sensitive data from the study participants, we ask the editors to consider the following statement to be written in the manuscript, if accepted for publication: 

Data Availability: Data underlying the study cannot be made publicly available due to ethical concerns, as data contain several personally identifiable information. Data are available from Oswaldo Cruz Foundation for researchers who meet the criteria for access to confidential data. Contact information: Institutional Ethics and Research Committee of the Evandro Chagas National Institute of Infectious Diseases, email: cep@ini.fiocruz.br or Guilherme Amaral Calvet; email: guilherme.calvet@ini.fiocruz.br

We note that you have included the phrase “data not shown” in your manuscript. Unfortunately, this does not meet our data sharing requirements. PLOS does not permit references to inaccessible data. We require that authors provide all relevant data within the paper, Supporting Information files, or in an acceptable, public repository. Please add a citation to support this phrase or upload the data that corresponds with these findings to a stable repository (such as Figshare or Dryad) and provide and URLs, DOIs, or accession numbers that may be used to access these data. Or, if the data are not a core part of the research being presented in your study, we ask that you remove the phrase that refers to these data.

This is an important point, thank you. We agree with the editor and have deleted this sentence because the data are not a core part of the present research being presented. 

REVIEWER 1 

The manuscript has stated objectives:

1) To assess the exposure, infection, and persistence of SARS-CoV-2 in nasopharyngeal and oropharyngeal secretions (nasal and oral swabs) and feces (rectal swab) of dogs and cats living in households with a COVID-19 human case.

2) To investigate clinical and laboratory alterations (complete blood count and serum biochemistry) in pets associated with the SARS-CoV-2 infection.

3) To evaluate risk factors associated with SARS-CoV-2 infection in pets.

However, the objective 2 does not appear to be pertinent to the context of the manuscript and its results were not really discussed. I recommend the second objective and related parts of the manuscript be removed.

We agree with the reviewer and remove the objective 2 from the text. But we have decided to keep the results in the article. So far, few published articles provide complete blood count and biochemistry results. All pets performed these tests at three different times. We believe that maintaining this data adds information to the study.

144, 210, 212, 414 etc.: “… human’s samples” should be “… human samples” Thank you for your pertinent suggestion. We have changed human’s samples” to “human samples”. Lines: 139, 193, 195.

145-147: “All VTM samples had viral RNA extracted manually by the QIAamp Viral RNA

Mini kit (Qiagen), or automatedly by Perkin-Elmer Chemagic machine/chemistry…” should be re-phrased because QIAamp Viral RNA kit actually extracts both DNA and RNA, that is total nucleic acids (no DNase treatment). If Perkin-Elmer Chemagic does not include DNase than it would be appropriate to say:

“Nucleic acids were extracted from VTM samples by QIAamp Viral RNA Mini kit (Qiagen), or by Chemagic instrument (Perkin-Elmer)…”

Also please specify the instrument model – was it Chemagic Prime, Chemagic 360 or Chemagic Prepito?

Thank you for this relevant recommendation. We agree with the reviewer’s points and have corrected the text as suggested.

New sentence: Nucleic acids were extracted from VTM samples by QIAamp Viral RNA Mini kit (Qiagen), or by Chemagic 360 instrument (Perkin-Elmer). Lines: 140-141.

150: “Amplifications were conducted …” should be changed to “Reverse transcription and amplification were conducted…”

Thank you for this very appropriate suggestion. We have included “Reverse transcription and” in the sentence. Lines: 143-144.

213-215: “…tested by a RT-PCR designed to amplify N1 and N2 genes of SARS-CoV-2, as previously described [22]. Pet samples were only considered positive for SARS-CoV-2 when at least two genes were amplified.” 

This should be re-phrased because: (i) it does not match the reference which describes detection of RdRp, E and N genes by real-time PCR assays; (ii) SARS-CoV-2 does not have N1 and N2 genes.

We have changed the sentence and included a new reference to support the statement made in the text.

"...tested by a RT-PCR designed to amplify two targets of gene N of SARS-CoV-2, as previously described [23] 

23. Centers for Disease Control and Prevention. 2019-Novel coronavirus (2019- nCoV) real-time rRT-PCR panel primers and probes. [Updated June 6 2020]. Available at: https://www.cdc.gov/coronavirus/2019-ncov/lab/rt-pcr-panel-primer-probes.html. Accessed March 12, 2021. Lines: 195-197.

244-145 “The PRNT is a highly specific serologic test and can be carried out on samples to

confirm the presence of neutralizing antibodies to SARS-CoV-2 [24].” Should be re-phrased because stating “to confirm the presence” implies a screening test was done before, therefore it is more appropriate to state” The PRNT is a highly specific serologic test for detection of neutralizing antibodies to SARS-CoV-2 [24].” 

We agree with this pertinent comment. We have changed the sentence to: “The PRNT is a highly specific serologic test for detection of neutralizing antibodies to SARS-CoV-2.” Lines: 227-228.

252: research facility of biosafety level 3 platform of IOC/Fiocruz. – define “IOC”

Thank you for pointing this out. We have changed the sentence to: “… all manipulation of infectious SARS-CoV-2 was performed in a multi-user research facility of biosafety level 3 platform of Oswaldo Cruz Institute (IOC)/Fiocruz. Lines: 232-235.

REVIEWER 2 

The material and methods section was extensively written, Please reduce it.

Thank you for this recommendation. We have decided to delete some parts of the text and rephrase others to reduce the section.

- Have you considered sampling dogs and cat from nearby households with negative SARS-CoV-2 owners , as a mean to add a control element to the study.

Thank you for your comment. But we haven't considered sampling dogs and cats from nearby households with negative SARS-CoV-2 owners. Our multidisciplinary project was developed in partnership with the Acute Febrile Illnesses Laboratory, and the Laboratory of Clinical Research on Dermatozoonoses in Domestic Animals, basing the collections of the animals after the identification of the human clinical cases that are being monitored. Not applicable

- Line 58: replace (infected/exposed) with (positive)

- In the entire text replace the word “exposed” and use the world “ infected or positive” because the demarcation between the two words is not sufficiently clear in the current prospective.

Thanks for the comment. We clarify that we used the word "exposed" to designate the presence of only neutralizing antibodies since there was no direct diagnosis with the presence of RNA or viral isolation. The presence of these antibodies indicates only previous exposure to the SARS-CoV-2. But we replace the term "exposed" with "seropositive" in the entire text for a better understanding.

We adopted the use of the term "infected" as defined by The World Organization for Animal Health (OIE) definition criteria for confirmed cases (infection) of SARS-CoV-2 in animals.

Lines: 58, 62, 238, 248, 331, 346, 350, 445, 448, 452, 455, 460, 560, 580. 

-Line 342: the use of plasma in serological testing is not recommended because of the possibility of nonspecific result. Please discard, if any, serological results based on the use of plasma.

Different biological samples are used to search for neutralizing antibodies, including serum, plasma, and cerebrospinal fluid. All of them undergo inactivation for 56 to 30 minutes to destroy proteins in the complement system that could influence neutralization. Besides, none of the tested plasma samples were seropositive in the study, and therefore there was no unspecific neutralization of the tested plasma samples. We have added the number of serum (n=92) and plasma (n=8) samples tested in the study. Lines: 324-325.

Line 514 this sentence should be removed “It was also experimentally demonstrated that cats can become infected and 515 transmit SARS-CoV-2 to other cats” because of its poor relation to the rest of the paragraph. Thank you for the insightful comment. We have deleted the sentence. 

REVIEWER 3 

1- One of the major concerns is the lack of description and comparison of the obtained sequences from animals to that of the circulating strains in humans at the time of their longitudinal study. It is highly recommended to sequence the full-length genome or at least the full-length S gene sequences of some isolates from animals and compare it to that of human. Such information will help great impacts in improving the quality of this manuscript. Meanwhile, presenting theses sequences and making them publically available is of high demands.

We agree with this study's limitation and that we recognize and describe in lines 527 to 535 (original manuscript). Also, all sequences obtained in the animals sampled in this study are deposited in an open-access database (GISAID), and their access numbers are described in table two. 

2- The authors must map in more precise details the sampling schedules from both human and animals.

Thank you for this recommendation. We have detailed sampling schedules. Lines: 130-132.

3- A detailed description of the outcomes of the viral infection in animals in correlation to human must be presented.

None of the cases of human patients diagnosed with COVID-19 needed hospitalization and the infection resolved itself with no signs of sequelae. Also, there was no need for specific medical care during the follow-up appointments of the animals positive for SARS-CoV-2. No human or animal deaths were observed during the study. We have included this information in the text. Lines: 288-291.

4- The statistical analysis in the M&M is long. Please consider shortening this section and use key references instead.

Thank you for your suggestion. We have decided to delete some parts of the statistical analysis section.

5- The discussion section is very long. Some paragraphs are too vague. Please consider shortening theses sections and mainly focusing this section on discussing your findings in comparison to other international groups that have conduced similar studies especially from USA, Asia, and Europe.

Thank you for pointing this out. We agree with the reviewer and delete some sentences in the discussion section. 

6- Please consider deleting Objectiv-2 since it is not adding too much to the main theme of this manuscript. Thank you for your comment. We decided to delete the objective 2 in the introduction section of the manuscript, but we have decided to keep the results in the article as previously mentioned in the response to reviewer #1.

7- Please try to highlight the roles of the One Health concept as an important approach for studying this type of research and its potential roles in the mitigation and control of the emerging and re-emerging diseases.

Thank you for this relevant recommendation. We have included the following paragraph in the discussion section:

“The present study highlights the role of the One Health approach in the mitigation and control of the COVID-19 pandemic [44, 45]. The integrated surveillance infrastructure of the Evandro Chagas National Institute of Infectious Diseases and the Regional Reference Laboratory in the Americas for Coronavirus was able to timely monitor and detect the occurrence of SARS-CoV-2 infection in both humans and animals using a multi-professional approach, including veterinarians, physicians, other health professionals, and a statistician The results of our study have been provided to The Brazilian Ministry of Agriculture, Livestock and Food Supply and can contribute to the investigation of animal SARS-CoV-2 hot spots. Lines: 515-523.

---

## [Decision Letter · Decision Letter 1]

15 Apr 2021

Investigation of SARS-CoV-2 Infection in Dogs and Cats of Humans Diagnosed with COVID-19 in Rio de Janeiro, Brazil

PONE-D-21-02395R1

Dear Dr. Calvet,

We’re pleased to inform you that your manuscript has been judged scientifically suitable for publication and will be formally accepted for publication once it meets all outstanding technical requirements.

Kind regards,

Maged Gomaa Hemida, ph.D

Academic Editor

PLOS ONE

Additional Editor Comments (optional):

Dear authors, I am pleased to let you know that your manuscript is greatly improved after a rigor round of revision and now is accepted for publication in PLos One

Reviewers' comments:

Reviewer's Responses to Questions

**Comments to the Author**

1. If the authors have adequately addressed your comments raised in a previous round of review and you feel that this manuscript is now acceptable for publication, you may indicate that here to bypass the “Comments to the Author” section, enter your conflict of interest statement in the “Confidential to Editor” section, and submit your "Accept" recommendation.

Reviewer #2: All comments have been addressed

Reviewer #3: All comments have been addressed

2. Is the manuscript technically sound, and do the data support the conclusions?

Reviewer #2: Partly

Reviewer #3: Yes

3. Has the statistical analysis been performed appropriately and rigorously? 

Reviewer #2: Yes

Reviewer #3: N/A

4. Have the authors made all data underlying the findings in their manuscript fully available?

Reviewer #2: Yes

Reviewer #3: Yes

5. Is the manuscript presented in an intelligible fashion and written in standard English?

Reviewer #2: Yes

Reviewer #3: Yes

6. Review Comments to the Author

Reviewer #2: the anthers adequately addressed my comments raised in a previous round. Generally. this manuscript met the criteria of well designed and conducted study,

Reviewer #3: Although the authors did not show any virus sequence from human came in close contact with positive SARS-CoV2

animals, they listed this point as one of the major limitation oh the current study. The revised manuscript is greatly improved

7. PLOS authors have the option to publish the peer review history of their article (what does this mean?). If published, this will include your full peer review and any attached files.

Reviewer #2: No

Reviewer #3: **Yes: **Maged Gomaa Hemdia

---

## [Editor Report · Acceptance letter]

16 Apr 2021

PONE-D-21-02395R1 

Investigation of SARS-CoV-2 Infection in Dogs and Cats of Humans Diagnosed with COVID-19 in Rio de Janeiro, Brazil 

Dear Dr. Calvet:

I'm pleased to inform you that your manuscript has been deemed suitable for publication in PLOS ONE. Congratulations! Your manuscript is now with our production department. 

Kind regards, 

on behalf of

Dr. Maged Gomaa Hemida 

Academic Editor

PLOS ONE